# Wnt-3a Induces Epigenetic Remodeling in Human Dental Pulp Stem Cells

**DOI:** 10.3390/cells9030652

**Published:** 2020-03-07

**Authors:** Verónica Uribe-Etxebarria, Patricia García-Gallastegui, Miguel Pérez-Garrastachu, María Casado-Andrés, Igor Irastorza, Fernando Unda, Gaskon Ibarretxe, Nerea Subirán

**Affiliations:** 1Cell Biology and Histology Department, University of the Basque Country (UPV/EHU), Barrio Sarriena, S/N, 48940 Leioa, Spain; vero18791@gmail.com (V.U.-E.); patricia.garcia@ehu.eus (P.G.-G.); mperez282@gmail.com (M.P.-G.); mdcasado002@gmail.com (M.C.-A.); iirastorza004@gmail.com (I.I.); fernando.unda@ehu.eus (F.U.); 2Pathology Department, New York University, 550 1st Avenue, New York, NY 10016, USA; 3Unité Mixte de Recherche UMR1029. INSERM-Université de Bordeaux, 33000 Bordeaux, France; 4Physiology Department, University of the Basque Country (UPV/EHU), Barrio Sarriena, S/N, 48940 Leioa, Spain; nerea.subiran@ehu.eus

**Keywords:** dental pulp stem cells, chromatin remodeling, cell cycle, pluripotency, DNA methylation, histone acetylation, histone methylation, Notch pathway, Wnt pathway

## Abstract

Dental pulp stem cells (DPSCs) from adult teeth show the expression of a very complete repertoire of stem pluripotency core factors and a high plasticity for cell reprogramming. Canonical Wnt and Notch signaling pathways regulate stemness and the expression of pluripotency core factors in DPSCs, and even very short-term (48 h) activations of the Wnt pathway induce a profound remodeling of DPSCs at the physiologic and metabolic levels. In this work, DPSC cultures were exposed to treatments modulating Notch and Wnt signaling, and also induced to differentiate to osteo/adipocytes. DNA methylation, histone acetylation, histone methylation, and core factor expression levels where assessed by mass spectroscopy, Western blot, and qPCR. A short-term activation of Wnt signaling by WNT-3A induced a genomic DNA demethylation, and increased histone acetylation and histone methylation in DPSCs. The efficiency of cell reprogramming methods relies on the ability to surpass the epigenetic barrier, which determines cell lineage specificity. This study brings important information about the regulation of the epigenetic barrier by Wnt signaling in DPSCs, which could contribute to the development of safer and less aggressive reprogramming methodologies with a view to cell therapy.

## 1. Introduction

Dental pulp stem cells or DPSCs arise from the neural crest (NC) as many other cells of craniomaxillofacial tissues [1,2,3,4,5,6,7]. Interestingly, DPSCs present some advantages with respect to other multipotent stem cell populations found in the adult human body [8,9,10], and they show a relatively high expression of core pluripotency factors like OCT4, SOX2, KLF4, LIN28, SSEA1, and NANOG [2,9,10,11,12,13]. The expression of those core factors regulates stem cell pluripotency [14,15,16,17,18]. The use of DPSCs could also be very relevant to cell therapy because these cells are also known to be easily accessible for extraction under aseptic conditions, well-tolerated upon grafting due to their immune-suppressive properties [19], non-tumorigenicity [20], and suitability for autologous therapy [7,21,22]**.**

Classical experiments showed that somatic cells could be reprogrammed into a pluripotent state by the ectopic expression of just a few core factors. The first recipe ever published featured the so-called Yamanaka factors: OCT4, SOX2, KLF4, and C-MYC [23]. Later on, other combinations of core factors with reprogramming effect were characterized, like: OCT4, SOX2, NANOG, and LIN28A [16]. However, the efficiency of these and other related reprogramming methods is usually very low (<1%), because of the restrictions imposed by the epigenetic determination barrier, which carries with it all the history of the starting parental adult cell [23,24]. Cell reprogramming and the induction of pluripotency depend critically on the erasure of the epigenetic tags linked to cell differentiation. Therefore, the study of these multiple epigenetic modifications associated with the differentiation process, and how these could be reversed, is of paramount importance. Some authors have demonstrated the superior ability of DPSCs to undergo full cell reprogramming. This was associated to a relatively non-methylated state of the DPSC genome in some critical loci, showing a fairly similar methylation pattern to that found in pluripotent stem cells (PSCs) such as embryonic stem cells (ESCs) and induced pluripotent stem cells (iPSCs) [25]. Recent studies also showed that inhibition of DNA methylation enhanced the reprogramming efficiency of gingival mesenchymal stem cells (hGMSCs; a population derived from neural crest as DPSCs), into embryoid body forming PSCs, suggesting the possible application of these approaches in autologous cell therapy and organ repair [26].

The epigenome of stem cells is different from the genome of differentiated cells. Stem cells present a genome predominantly in euchromatic conformation whereas the genome of somatic cells is more enriched in heterochromatin, with a higher amount of genes permanently silenced by cytosine methylation [27,28,29]. It is also known that DNA methylation levels are low in PSCs both in vitro and in vivo [24,30]. Additionally, a high level of histone acetylation in PSCs is known to contribute to the weakening of the interaction with DNA, leading to the unfolding of chromatin and activation of gene transcription [31]. Contrarily, the loss of acetylation catalyzed by histone deacetylases (HDACs) leads to a closed heterochromatin conformation, thereby repressing transcriptional activity [32]. Chromatin remodeling in PSCs is also regulated by histone methylation. The most characterized are the tri-methylation marks of Histone 3, and the general description attributes to H3K4me3 a role as a gene activator mark, whereas H3K9me3 and H3K27me3 are known as repressive marks associated with transcriptional silencing [33]. In fact, a distinctive characteristic of PSCs is the presence of bivalent domains containing both activating (H3K4me3) and repressing (H3K27me3) histone methylation marks in genes governing cell stemness and differentiation [34,35,36,37]. These bivalent domain-containing genes would enable the stem cell to respond to changes in the environment by rapidly activating the transcription of specific sets of genes while repressing others [35,38].

The canonical Notch and Wnt signaling pathways are widely regarded as important regulators of stemness [39,40,41,42,43] and cell differentiation [44,45,46] in DPSCs and many other stem cell types. It is known that dental stem cells present particularly high levels of Wnt and Notch pathway activity, which relates to their high expression of pluripotency core factors and high ability for cell reprogramming [2,9,13,47,48]. However, the epigenetic regulations that these pathways may exert on DPSCs are still unclear. A better understanding of how Notch and Wnt influence the epigenetic tags in DPSCs could lead to significant improvements in the efficiency of the current nuclear reprogramming methodologies using these cells, and in the ex vivo expansion of stem and differentiated cells for cell transplant therapies. For this purpose, we studied the epigenetic profile of DPSCs treated with Notch and Wnt signaling pharmacological modulators and we assessed the similarities and differences in the DNA methylation, histone methylation and histone acetylation patterns comparing to control DPSCs, DPSCs exposed to osteogenic and adipogenic differentiation conditions, and PSCs.

## 2. Materials and Methods

### 2.1. DPSC Culture

DPSCs were isolated from human third molars obtained from healthy donors between 18 and 30 years of age, who gave their informed consent for donation, following the approval of the CEISH committee of UPV/EHU for research with human samples, and abiding by the ethical principles of the Declaration of Helsinki on medical research involving human subjects. Teeth were fractured and enzymatic digestion of the pulp tissue was carried out for 1 h at 37 °C with 3 mg/mL collagenase (17018-029, Thermo Fisher Scientific, Waltham, MA, USA) and 4 mg/mL dispase (17105-041, Thermo Fisher Scientific), followed by mechanical dissociation. The DPSCs were cultured in Dulbecco’s modified Eagle’s medium (DMEM) supplemented with 10% fetal bovine serum (FBS), L-glutamine (1 mM) and the antibiotics penicillin (100 U/mL) and streptomycin (150 µg/mL). After 1 week in culture, practically 100% of cells were positive for the neural markers Nestin, β3-tubulin, the mesenchymal markers Collagen I, CD90, CD105, CD73, and negative for the haematopoietic marker CD45 [49]. The DPSCs could be amplified and maintained in these conditions for very long periods (>6 months). However, to avoid cell aging issues, we only employed DPSCs that had been grown in culture for less than 3 months and had accumulated no more than 6 total passages. Comparative experiments between control and treatment conditions were always and without exception performed in parallel using DPSCs from the same donor.

### 2.2. ESC Culture

Mouse Oct4-GFP ES Cells (PCEMM08; PrimCells LLC, San Diego, CA, USA) were cultured in 2i+LIF feeder-free culture conditions on a dish coated with gelatin (0.01%) in Knockout DMEM medium (Knockout serum replacement 15%, sodium pyruvate 1mM, non-essential aminoacids supplement 1% (M7145, Sigma, San Luis, MO, USA), penicillin/streptomicin 100 U/mL, L-glutamine 1% and β-mercaptoethanol 0.007% with 2i (PD0325901, 0.4 mM; Stemgent, San Diego, CA, USA, and CHIR99021, 3 mM; Stemgent, San Diego, CA, USA) and LIF(1000U/mL; Sigma, San Luis, MO, USA). The expanded ESC colonies were passaged by dissociation with TrypLE (Invitrogen, Carlsbad, CA, USA).

### 2.3. Notch and Wnt Pathway Pharmacological Modulation

To block Notch signaling, we employed DAPT ((N-[N (3, 5-diflorophenacetyl-L-alanyl)] 5-phnylglycine t-butyl ester), a γ-secretase inhibitor, (565784, Calbiochem, San Diego, California, CA, USA), at a concentration of 2.5 µM. DAPT was added to the culture medium for 48 h prior to the assays where DAPT-treated DPSCs were compared with DPSCs treated only with the control vehicle DMSO. To activate Wnt signaling, we used 2.5 µM BIO (6-bromoindirubin-3′-oxine), a GSK3β inhibitor (361550, Calbiochem, San Diego, CA, USA), which was added to the medium for 48 h prior to the assays. BIO-treated cells were compared with DPSCs exposed to the inactive analog MBIO (methyl-6-bromoindirubin-3′-oxine) at 2.5 µM as a corresponding control (361556, Calbiochem). WNT-3A recombinant protein (5036-WN-010, R&D Systems, Minneapolis, MN, USA) was diluted in PBS and also added at 2.5 µM for 48 h to the DPSCs cultures as another treatment to activate Wnt signaling.

### 2.4. Osteogenic Differentiation of DPSCs

We used the following protocol to induce DPSC differentiation to mature osteoblasts: 6 µM β-glycerolphosphate (G9422, Sigma-Aldrich, St. Luis, MO, USA), 10 nM dexamethasone (D4902, Sigma, San Luis, MO, USA), and 52 nM ascorbic acid (127.0250, Merck, Darmstadt, Germany) were added to the cell cultures in DMEM + 10% FBS for three weeks prior to the assays. Osteoblastic differentiation pre-commitment was assessed by Alkaline Phosphatase (ALP) staining in control and treated cells. Cells were fixed for 1min using 4% paraformaldehyde and washed with 0.05% Tween 20 in PBS. ALP staining was performed using 5-Bromo-4-chloro-3-indolyl phosphate/Nitro Blue tetrazolium (BCIP/NBT; Sigma, San Luis, MO, USA) as chromogen substrate, and the staining progress was checked every 3 min. After the incubation, cells were washed three times with PBS and ALP absorbance at 405 nm was quantified using a Synergy HT Multi-Mode Microplate Reader (Biotek, Winooski, VT, USA). Terminal differentiation to mature mineralizing osteoblast/osteocyte lineage cells was assessed at 3 weeks post-induction by detection of extracellular calcified bone matrix deposits via Alizarin Red staining, using 2 g/100 mL Alizarin Red S (400480250, Across Organics, Geel, Belgium) at pH 4.3. DPSCs were fixed with 10% formalin (F7503, Sigma, San Luis, MO, USA) for 30 min, incubated with the Alizarin S Red solution for 45 min, and washed four times with PBS to remove any background staining. Alizarin Red absorbance at 450 nm was quantified using a Synergy HT Multi-Mode Microplate Reader (Biotek, Winooski, VT, USA).

### 2.5. Adipogenic Differentiation of DPSCs

To induce adipogenic differentiation, we treated DPSC cultures with 0.5 mM IBMX (I5879, Sigma, San Luis, USA), 1 µg/mL insulin (91077C, SAFC Biosciences, St. Luis, MO, USA) and 1 µM dexamethasone (D4902, Sigma, San Luis, MO, USA) for four weeks prior to the assays. Terminal differentiation to adipocytes was assessed by Oil Red staining. Cells were fixed with 10% formalin for 10 min and then washed with PBS containing 60% Isopropanol. Lipid droplets in mature adipocytes were detected by incubation with a solution containing 5.14 µM Oil Red Stock (O-0625, Sigma, San Luis, MO, USA) in miliQ water for 10 min. Oil Red absorbance was measured at 490 nm using a Synergy HT Multi-Mode Microplate Reader (Biotek, Winooski, VT, USA).

### 2.6. RNA Extraction, Reverse Transcription and Quantitative Real-Time PCR (qPCR)

Cell pellets were frozen and stored at −80 °C. Total RNA was extracted from the cells using the RNeasy Kit (74104, Qiagen, Hilden, Germany) and checked for purity by measuring the 260/280 ratio in the Nanodrop Synergy HT instrument (Biotek, Winooski, VT, USA). cDNA (50 ng/µL) was obtained by reverse transcription of total extracted RNA using the iScript cDNA Kit (1708890, BioRad, Hercules, CA, USA) with the following reagents: iScript reverse Transcriptase (1 µL), 5× iScript Reaction Mix (4µL) and Nuclease Free water (variable) to a final volume of 20 µL. Quantitative Real-Time PCR experiments were conducted in an iCycler My iQ^TM^ Single-Color Real-Time PCR Detection System (BioRad, Hercules, CA, USA), using 4.5 µL of Power SYBR^®^ Green PCR Master Mix 2× (4367659, Applied Biosystems^TM^Applied Biosystems, Carlsbad, CA, USA), 0.5 µL of primers (0.3125 µM), 0.3 µL of cDNA (1.5 ng/µL) and Nuclease Free water for a total volume reaction of 10 µL. All primers were obtained from public databases and checked for optimal efficiency (>90%) in the qPCR reaction under our experimental conditions. The relative expression of each gene was calculated using the standard 2^-ΔCt^ method [50] normalized with respect to the average of *β-ACTIN* and *GAPDH* as internal housekeeping control genes. All reactions were performed in triplicate. qPCR was run on a CFX96^®^ thermo cycler (BioRad, Hercules, CA, USA). Data were processed by CFX Manager™ Software (BioRad, Hercules, CA, USA). We assessed that all qPCR reactions yielded only one amplification product by the melting curve method. We used the following primer pairs for different human and mouse gene transcripts obtained via Primer Bank and validated by the NCBI Primer-Blast method (Table 1).

### 2.7. Protein Extraction

DPSCs were washed with hand-warm PBS several times, and the proteins were lysed on ice with 200 µl Lysis Buffer (50 mM Tris-HCl pH 7.5, 1mM EDTA, 150 mM NaCl, 0.5% sodium deoxycholate, 0.1% SDS, 1% IGEPAL^®^ CA-630 in dH_2_O), Proteinase Inhibition (1:100, 539134, Calbiochem) and phosphatase inhibitor cocktail (1:100, 539134, Calbiochem, San Diego, CA, USA). After an incubation of 5 min lysates were scrapped thoroughly and transferred to a pre-chilled 1.5 mL tube and submitted to homogenization in a Bandelin Plus sonicator using a 1.5 MS probe. After three sonication bursts on ice for 20 s at 90% amplitude with 1 min of rest in between each of them, lysates were cleared by centrifugation at 20,000 rcf for 10 min at 4 °C. Supernatants were quantified using CuSO_4_-BCA in 50:1 ratio (B9643, Sigma, San Luis, MO, USA), BSA (A7906, Sigma, San Luis, MO, USA) was used for performing a linear relationship with concentration and absorbance at 490nm. Samples were read in the Nanodrop Synergy HT (Biotek, Winooski, VT, USA).

### 2.8. Western Blot (WB)

The samples with 30 µg of total protein were diluted in loading buffer (62.5 mM Tris-HCl, pH 6.8, 2.5% SDS; 10% glycerol; 5% β-mercaptoethanol and 0.002% bromophenol blue). After electrophoretic separation (electrophoresis buffer formulation: 25 mM Tris, pH 8.3; 193 mM glycine, 0.1% SDS) under constant 120 V, proteins were blotted during 3 h (250 V max, 600 W max, constant 400 mA) using Transfer buffer: 25 mM, Tris pH 8.3; 192 mM glycine; 20% methanol; 0.1% SDS) onto 0.2 µm-pore nitrocellulose membranes using the Mini-PROTEAN tetra system and Mini Trans-Blot cell respectively fed by a PowerPac HV^TM^ High-Current Power Supply. Once correct protein transfer was confirmed by Ponceau S protein staining, membranes were washed with TBST (10 Mm Tris-HCl, pH 8; 150 mM NaCl; 0.05% Tween 20) until all dye was gone and submitted to blockage using 1% BSA diluted in TBST during 1 h at room temperature under constant agitation. For Western blot analyses, we used anti α-TUBULIN (1:3000, 4967, Cell Signaling, Massachusetts, MA, USA), anti-H3K9me3 (1:2000, 9542S, Cell Signaling, Massachusetts, MA, USA), anti-H3K4me3 (1:1000, ab12209, Abcam, Massachusetts, MA, USA), anti-H3K27me3 (1:1000, ab6002, Abcam, Massachusetts, MA, USA), anti-H3AC (1:2000, 06–599, Millipore) anti-DNMT1 (1:2000, 60B1220.1, Novus biologicals, Littleton, CO, USA), anti-DNMT3A (1:1000, 64B1446, Novus biologicals, Littleton, CO, USA), anti-DNMT3B (1:500, orb229237, Biorbyt, Cambridge, UK), and anti-CYCLIND (1:1000; 92G2, Cell Signaling, Massachusetts, MA, USA). The secondary antibodies used were: mouse IgGκ light chain binding protein HRP conjugated 1:5000 (Santa Cruz sc-516102), anti-rabbit-HRP 1:5000 (Santa Cruz sc-2357), and anti-rat-HRP 1:4000 (Santa Cruz sc-2006, Dallas, TX, USA). The blots were developed using the Luminata Crescendo Western HRP Substrate (WBLUR0500 Millipore, Burlington, MA, USA). Western blot images were taken in a Syngene G: BOX CHEMI XR^5^system (Syngene, Cambridge, UK). The membranes were stripped using Red Blot (M2504, Inmmobilon^®^ EMD Millipore, Burlington, MA, USA). Samples were quantified by Fiji-ImageJ [51] after background subtraction.

### 2.9. DNA Extraction

Cell lysates were made using DNA lysis Buffer (100 mM Tris-HCl, 50 mM EDTA, 200 mM NaCl and 0.2% SDS) and Proteinase K (AM2546, Thermo Fisher Scientific Waltham, MA, USA) which was used at 100 mg/mL and samples were incubated overnight under gentle shaking. The samples were treated with 5 µL RNAse at 10 mg/mL (800-325-3010, Roche, Basilea, Switzerland), during 1 h at 37 °C. DNA extraction was performed using a classical phenol-chloroform methodology with phenol (P1037, Sigma) and chloroform reagents (288306, Sigma). After the extraction, the DNA concentration and purity were checked by measuring the 260/280 absorbance ratio in the Nanodrop Synergy HT (Biotek, Winooski, VT, USA).

### 2.10. Quantification of DNA Methylation by Mass Spectroscopy (MS)

The extracted DNA was enzymatically hydrolyzed and the aliquoted samples (10 μL typically containing 50 ng of digested DNA) were run in a reverse phase UPLC column (Eclipse C18 2.1 × 50 mm, 1.8 μm particle size, Agilent, Santa Clara, CA, USA) equilibrated and eluted (100 μL/min) with water/methanol/formic acid (95/5/0.1, all by volume). The effluent from the column was added to an electrospray ion source (Agilent Jet Stream) connected to a triple quadrupole mass spectrometer (Agilent 6460 QQQ, Santa Clara, CA, USA). The machine was operated in the positive ion multiple reaction monitoring mode using previously optimized conditions, and the intensity of specific MH+→fragment ion transitions were measured and recorded (5 mC *m*/*z* 242.1→126.1, 5hC 258.1→142.1 and dC *m*/*z* 228.1→112.1). The measured percentage of 5 mC in each experimental sample was calculated from the MRM peak area divided by the combined peak areas for 5 mC plus 5hmC plus C (total cytosine pool).

### 2.11. Cell Cycle Phase Determination

Cells were trypsinized and diluted in suspension in 100% ethanol. Determination of cell cycle phase was assessed by flow cytometry using 0.5 mg/mL Propidium Iodide (P4170, Sigma, San Luis, MO, USA) and 10 µg/mL Ribonuclease RNAse (R4642, Sigma, San Luis, MO, USA). Samples were read using CytoFLEX Flow Cytometer (Beckman Coulter, Brea, CA, USA) and analyzed with Kaluza G for Gallios Acquisition Software (Beckman Coulter, Brea, CA, USA).

### 2.12. Statistical Analyses

Statistical analyses were performed with Excel, IBM SPSS Statistics v.9 (SPSS, Chicago, IL, USA) and Graph Pad v.6 software (Graph Pad Inc., San Diego, CA, USA). We used non-parametric statistical tests to compare the different control and treatment conditions. Comparisons between only two groups were made using U-Mann Whitney test. Comparisons between multiple groups were made using Kruskal–Wallis followed by Dunn´s post hoc test. *p* ≤ 0.05 was considered statistically significant.

## 3. Results

### 3.1. Wnt Activity Reverses Osteogenic Cell Differentiation and Increases the Expression of Core Pluripotency Factors in DPSCs

DPSCs were cultured in DMSO (control), DAPT, MBIO (control), BIO, and WNT-3A treatment conditions for 48 h. When grown in standard medium containing 10% FBS, DPSCs tend to spontaneously differentiate to mineralizing osteo/odontoblastic cell phenotypes [52,53]. Osteoblastic cell commitment was assessed by the detection of Alkaline Phosphatase (ALP) reaction in DPSC cultures. Interestingly, we found that the application of either BIO or WNT-3A significantly reduced ALP staining (Figure 1A,B), suggesting that Wnt activation could revert the default osteoblastic lineage pre-differentiation phenotype of DPSCs in standard culture conditions.

We also included other treatments to induce the terminal differentiation of DPSCs to mature cell lineages. In particular, control DPSCs were exposed to two well-established osteogenic and adipogenic differentiation induction media [5,54]. After an induction period of between three and four weeks, we assessed the terminal differentiation of DPSCs to osteoblasts and adipocytes, respectively. Osteoblastic differentiation was demonstrated by the detection of mineralized bone matrix deposits stained with Alizarin S Red after three weeks of induction (Figure 1C; bottom panel), together with an increased expression of mature osteoblastic gene markers SPARC (Osteonectin) and OSTERIX/SP7 (Appendix A). Terminal differentiation of DPSCs to adipocytes was assessed by the detection of abundant cytoplasmic lipid droplets stained with Oil Red (Figure 1C; top panel), together with an overexpression of adipocyte gene markers PPARγ and LPL (Appendix A). Adipocyte differentiation also came along with a notorious change in cellular morphology at four weeks of induction, where DPSCs lost their usual spindle-like shape and instead adopted a rounded cell appearance (Figure 1C; top panel).

Quantitative real time PCR analysis showed that terminally differentiated DPSCs underwent a consistent and significant decrease in transcript expression for the pluripotency core factors c-MYC, SOX2, OCT4 and NANOG, almost in all cases to less than half of basal control levels (Figure 1D). A similar effect was found when DPSCs were exposed to Notch inhibitors (DAPT) for 48 h. On the contrary, when DPSCs were exposed to Wnt activators WNT-3A or BIO for 48 h, they showed a consistent overexpression of all core factor genes. The ones most upregulated were SOX2 and OCT4A (Figure 1E). The upregulation of core factor expression went in parallel to a downregulation of osteoblastic and adipogenic differentiation markers in DPSCs (Appendix A). OSTERIX expression was downregulated to about half of control levels by either BIO or WNT-3A application, thus confirming a suppression of the default osteoblastic differentiation pathway in Wnt-activated DPSC cultures.

### 3.2. Notch and Wnt/β-Catenin Signaling Regulates the Cell Cycle in DPSCs

Notch and Wnt signaling were known to affect the proliferative ability of DPSCs [13]. To precisely evaluate the impact of these treatments over the cell cycle, the relative rates of G0/G1, S and G2/M cell cycle phases by flow cytometry were studied after exposure of DPSCs to DAPT, BIO and WNT-3A. We observed significant differences in the cases of BIO and WNT-3A treatments with respect to controls, with a higher proportion of cells in S-G2/M phases (20.3%, 23.75%) and a lower proportion of cells in G0/G1 (Figure 2D,F,H). Additionally, we studied the expression of CYCLIN D1, a key regulator of the transition between G1-S and S-G2 phases of the cell cycle. CYCLIN D1 transcription decreased significantly in DAPT-treated DPSCs and was almost abolished in DPSCs subjected to osteogenic and adipogenic differentiation, to less than 10% of control levels (Figure 2I). Regarding Wnt activation treatments, BIO and WNT-3A increased CYCLIN D1 transcription in DPSCs (Figure 2J) but these changes did not translate into changes at protein level, as assessed by WB (Figure 2G).

### 3.3. WNT-3A Leads to a DNA Hypo-Methylation State in DPSCs

To investigate whether Notch and Wnt signaling would also control the epigenetic profile of DPSCs, the global DNA methylation pattern of DPSC cultures was studied by high resolution mass spectroscopy (MS). We observed a significantly higher proportion (%) of 5methyl-cytosine (5 mC) with respect to total cytosine in the genomic DNA of DPSCs both after the osteoinduction treatment (2.71 ± 0.14%: *p* < 0.01) and adipoinduction treatment (2.81 ± 0.06%: *p* < 0.01) with respect to control DPSCs (2.41 ± 0.05%; Figure 3A). Interestingly, DPSCs treated with WNT-3A significantly diminished their % of 5 mC content (2.20 ± 0.05%: *p* < 0.05) whereas we did not find significant changes in BIO (2.66 ± 0.06%) with respect to either control DMSO or MBIO (Figure 3B). Mouse ESCs (mESCs) in a state of naïve pluripotency were included here to calibrate the system as a control of cells with very low levels of DNA methylation [55,56,57,58]. In a previous characterization, we corroborated that these cells presented comparatively higher levels of core factor expression than control DPSCs (Appendix A). As expected, we found that mESCs also presented significantly lower DNA methylation levels, as assessed by MS (1.34 ± 0.02%; Figure 3B).

In view of these results, we wondered if the decreased DNA methylation levels observed in WNT-3A treated DPSCs could somehow be the consequence of alterations in the expression DNA methyltransferases. The expression of the different DNA cytosine-5-methyltransferases (DNMT) was assessed by WB. We found that the maintenance methyltransferase DNMT1 was the only one that could be reliably detected at protein level in DPSCs (Figure 3C). De novo methyltransferases DNMT3A and DNMT3B were not detected in DPSCs by WB, although they could be detected in the positive control of mESCs, especially in the case of DNMT3A (Figure 3C; right panel). However, the qPCR analysis determined that control DPSCs presented very small but nevertheless detectable transcript levels for all three DNA cytosine-5-methyltransferases 1, 3A, and 3B (DNMT1, DNMT3A and DNMT3B). The transcript expression levels were in all cases much higher for DNMT1 than for DNMT3A and DNMT3B, which were both marginally expressed (data not shown). Interestingly, the expression of DNMT1, DNMT3A and DNMT3B increased when DPSCs were exposed to osteoinduction and adipoinduction treatments, and to the Notch inhibitor DAPT (Figure 4A). On the contrary, when we exposed DPSC cultures to BIO or WNT-3A for 48 h, DNMT3A and DNMT3B transcript levels increased, but DNMT1 levels were not significantly affected (Figure 4B).

DNA methylation reactions depend on the availability of S-adenosylmethionine (SAM) substrate [59]. Cellular SAM is metabolized by the aforementioned DNMT enzymes, but also by other competing ones. One of the most representative SAM-consuming enzymes is the Nicotinamide N-methyltransferase or NNMT, which has been implicated in the generation of cellular methylation sinks [60]. We found that NNMT was particularly highly expressed at transcript level by DPSCs (25.8%, with respect to the housekeeping genes *β-ACTIN/GAPDH*: *p* < 0.05). Interestingly, we also found that NNMT expression was clearly downregulated in DAPT-treated and in differentiation-induced DPSCs, to less than half of basal control levels (Figure 4C), whereas it was upregulated in BIO and WNT-3A treated DPSCs (11.85 ± 3.23: *p* < 0.05; 2.34 ± 0.36: *p* < 0.05, respectively; Figure 4D).

### 3.4. Wnt Activation Increases Histone Acetylation in DPSCs

Another important epigenetic tag which determines stemness and cell differentiation is histone acetylation. Total protein expression levels of acetylated-Histone 3 (H3AC) were assessed by WB, and were found to be significantly higher in WNT-3A and BIO-treated DPSCs (126 ± 11.9%: *p* < 0.05; 146 ± 16.5%: *p* < 0.05, respectively) compared to normalized controls DMSO and MBIO (Figure 5A). The positive control of mESCs also showed higher levels of histone acetylation compared to control non-treated DPSCs (Figure 5A; right panel). The transcript expression levels for some known histone acetyltransferases (HATs) and histone deacetylases (HDACs) were also assessed by qPCR. DAPT-treated DPSCs and DPSCs which had been induced to differentiate to osteo/adipocytes had lower gene expression levels for the acetyl-transferase HAT/KAT8, and higher expression levels for the deacetylase HDAC/SIRT1, compared to control DPSCs (Figure 5B). On the contrary, when DPSCs were exposed to WNT-3A and BIO, the transcript expression of HAT/KAT8 increased significantly, whereas HDAC/SIRT1 expression was not affected (Figure 5C).

### 3.5. Wnt Activation Modifies the Histone H3 Methylation Pattern in DPSCs

In order to assess whether the pattern of H3 methylation was also affected by exposure of DPSCs to Notch and Wnt regulators, a WB analysis was performed against tri-methylated H3 Histones H3K4me3, H3K27me3 and H3K9me3. We found that protein expression levels were significantly increased for most of these three methylation tags in BIO, and especially, in WNT-3A treated DPSCs (H3K4me3: 133 ± 21,56%, *p* < 0.01; H3K27me3: 337 ± 67,22%, *p* < 0.05; and H3K9me3: 362 ± 27,5%, *p* < 0.05 respectively; Figure 6A), compared to their respective normalized controls. These results showed that Wnt activation changed the epigenetic footprint of DPSCs also at histone methylation level. Pluripotent mESCs also showed higher overall levels for all the three H3 methylation tags H3K4me3, H3K27me3, and H3K9me3, compared to control DPSCs (Figure 6A; right panel). Finally, we analyzed by qPCR the relative transcript expression for the key catalytic subunits responsible for each of these three H3 trimethylations: MLL (H3K4me3), EZH2 (H3K27me3), and EHMT2 (H3K9me3). We found that BIO and WNT-3A induced significant increases in the expression for MLL, EZH2, and EHMT2 in DPSCs. In contrast, terminal differentiation of DPSCs to osteocytes and adipocytes induced a downregulation of MLL, EZH2, and EHMT2 at the transcript level (Figure 6B,C).

## 4. Discussion

Traditional reprogramming methods of somatic cells rely on permanent genetic modification, which compromises the safety of these therapies for clinical use [61,62,63]. Hence, there is an amply justified interest in finding sources of human cells with a suitable epigenetic profile for a less invasive reprogramming. In this regard, DPSCs have emerged as some of the most promising candidates to cover the existent gap between research and the clinic. DPSCs show an exceptional ability for full cell reprogramming to iPSCs, even without a need for genome-integrating vectors [25].

Several studies have described canonical Notch and Wnt signaling pathways as pivotal regulators of stemness and pluripotency [40,64,65,66,67,68,69,70,71,72], and their pharmacological manipulation has already been tested as a strategy to enhance either cell differentiation [73] or cell reprogramming [13,74]. The pluripotency network comprises a core set transcription factors, including OCT4A (POUF51F), SOX2 and NANOG, which serve to establish the undifferentiated state and the self-renewal capacity of pluripotent stem cells [27]. This network interacts with the cell cycle machinery. The cell cycle of PSCs is characterized by a rapid progression and a minimal time spent in G1 [75,76,77]. In DPSCs treated with the Wnt activators BIO and WNT-3A, we did not find more than a 7% change in cells on G1/G0 phases of the cell cycle, compared to controls DMSO and MBIO. These changes were corroborated by numerous experiments and support the findings of previous studies [13] suggesting that these pharmacological treatments induced only a mild effect on DPSC proliferation and self-renewal.

DNA methylation is a critical regulator of stem cell differentiation. Interestingly, it has been described that the DNA methyltransferase inhibitor 5-aza-2′-deoxycytidine (5-aza-dC) is able to increase the stemness of periodontal ligament stem cells (PDLSCs), and also to enhance the reprogramming efficiency of human gingival mesenchymal stem cells (hGMSCs) [26,78]. Another report showed that a short pre-application of 5-aza-dC for 24 h increased the responsiveness of DPSCs to osteogenic differentiation protocols [79], a similar result to the one obtained after preconditioning DPSCs for 48 h with BIO or WNT-3A, as we showed in our own previous research [13]. Regarding other epigenetic marks such as acetylation, it was reported that p300, a well-known histone acetyltransferase, also played an important role in maintaining the stemness of DPSCs [80]. Moreover, different types of histone deacetylases or HDACs such as HDAC1, HDAC2, HDAC3, HDAC4, and HDAC9 act as important accelerators of odontoblast differentiation [81,82,83] and a similar effect has been described for histone demethylases like KDM6B, which removes H3K27me3 tags [84]. All of these evidences indicate an important regulatory circuit involving DNA and histone modifications that are crucial in determining the cell fate in DPSCs, and at least in some cases, these epigenetic regulations are linked to the Notch and Wnt signaling pathways, whose activities are mutually interdependent in these cells [13].

In our research model we induced DPSC cultures to generate terminally differentiated cells of osteocyte and adipocyte lineages. Pharmacological induction of DPSC differentiation increased DNA methylation with respect to control DPSC levels, corroborating the widely held view that somatic differentiated cells present a more methylated genome than stem cells [85]. Nevertheless, the most interesting findings of this work refer to the changes induced by Wnt/β-catenin signaling activation on the epigenetic profile of DPSCs. The recombinant protein WNT-3A, applied for just 48 h, induced a significant genomic DNA demethylation and a significant increase of global H3 acetylation in primary DPSC cultures. A genomic DNA methylation assessment by MS enabled us to establish a cell differentiation scale according to the levels of 5 mC, where terminally differentiated DPSCs to osteocytes and adipocytes had the highest levels of methylation, whereas naïve pluripotent PSCs showed the lowest. Based on these results, we think that MS could be a useful tool to monitor the global cell determination state of DPSCs and other stem cell cultures.

The principal DNA methyltransferase expressed by DPSCs was DNMT1. This maintenance methyltransferase only acts on hemimethylated DNA strands and is essential for the transmission of DNA methylation patterns to the daughter cells after DNA replication [86,87] In our model, we did not find significant changes in DNMT1 transcript and/or protein levels in DPSCs after Wnt activation. Regarding de novo methyltransferases DNMT3A and DNMT3B, their expression in DPSCs was very small and only found at the transcript level. Thus, it is unlikely that solely an altered expression of DNA methyltransferases could on its own account for the changes in genome methylation observed in DPSCs after WNT-3A exposure. It is more plausible that changes in DNA methyltransferase activity and/or DNA methylation turnover could be responsible for this effect [88] For instance, DNA methylation reactions require very large pools of available S-adenosylmethionine (SAM) as a substrate for the DNA methyltransferases, and thus these enzymes must compete with other SAM-consuming cellular methyltransferases like Nicotinamide-N-methyltransferase (NNMT), whose increased expression has been reported to generate methylation sinks [89]. Interestingly, the basal NNMT transcript expression levels in DPSCs were found to be characteristically high, and this expression was increased by several-fold under Wnt activation, thus providing a potential mechanism to explain the global decrease in DNA methylation levels observed in treated DPSCs. Other alternative mechanisms rely on the interplay between DNA methylation and histone acetylation [90].

In this study, increased levels of histone acetylation in DPSCs were found after exposure to WNT-3A. This has a direct connection with a recently described metabolic reprogramming induced by Wnt activation in these cells. Our previous research demonstrated that the treatment with WNT-3A exactly in the same conditions as in the present study activated both mitochondrial metabolism and lipid synthesis in DPSCs [91]. De novo cytoplasmic production of fatty acids requires a high availability of acetyl-coA, which comes primarily from citrate leaving the mitochondria in a process called cataplerosis [92]. Once in the cytosol this citrate is metabolized by the ATP-citrate lyase (ACLY) enzyme, also overexpressed in Wnt-activated DPSCs [91], which generates cytoplasmic and nuclear pools of acetyl-coA. The increased lipid synthesis observed in Wnt-activated DPSCs strongly suggests the possibility of cyto/nucleoplasmic acetyl-coA accumulation [91]. Importantly, high levels of acetyl-coA not only sustain lipid synthesis, but also histone acetylation, and this is known to be crucial for the maintenance of stemness and pluripotency [36,93]. Moreover, these stored lipids could also eventually become a reliable ready-to-use fuel for new acetyl-coA generation, thus sustaining high histone acetylation levels in the cell nucleus [91]. Thus, compelling evidence points to the coordinated activation of mitochondrial and lipid metabolism as a mechanism promoting an increased histone acetylation under Wnt activation in DPSCs. Finally, acetylation-induced transcription could also indirectly account for a global genomic DNA demethylation effect, as the latter has been proposed as a memory mechanism for active gene transcription [94].

Stem cells and particularly PSCs are also known to present different histone methylation patterns compared to somatic differentiated cells, where the most studied are trimethylation marks in H3 [38]. Interestingly, after WNT-3A exposure we found that the levels for both activating (H3K4me3) and repressing (H3K9me3, H3K27me3) H3 trimethylation tags were increased in DPSCs. These studies are in concordance with observations in ESCs where activating marks such as H3K4me3 are known to be combined with H3K27me3 repressive marks in lineage-specific genes [95]. Thus, it is tempting to argue that exposure to WNT-3A induces a chromatin remodeling in DPSCs, to approach PSC characteristics. More studies would be necessary to analyze whether these changes in H3 trimethylation are directly related to the emergence of bivalent gene promoters in DPSCs after Wnt activation.

## 5. Conclusions

In this article, we hypothesized that the metabolic remodeling induced in DPSCs by Wnt activation could also have an impact in the epigenetics of DPSCs. We found out that WNT-3A exposure induced multifaceted epigenetic reprogramming in DPSCs, characterized by a global DNA hypomethylation, a global histone hyperacetylation, and an increase in both activating and repressing histone methylation marks, which constitute the most typical epigenetic footprints of PSCs. These findings shed light on how stemness, signaling, metabolic, and epigenetic networks cooperate in DPSCs, and they could also have important implications to optimize the clinical use of these cells. The fact that a simple pharmacological treatment with WNT-3A can induce a chromatin remodeling in DPSCs brings important practical advantages with a view to cell therapy, such as the possibility to keep these cells for longer periods in culture in the presence of FBS, without compromising their stemness properties due to spontaneous in vitro osteoblastic differentiation. Another potential application could be to design better and more efficient protocols for the differentiation of DPSCs to many different lineages of somatic cells. Finally, this work could also contribute to the development of gentler, safer, and more efficient full cell reprogramming strategies using DPSCs.

## Figures and Tables

**Figure 1 cells-09-00652-f001:**
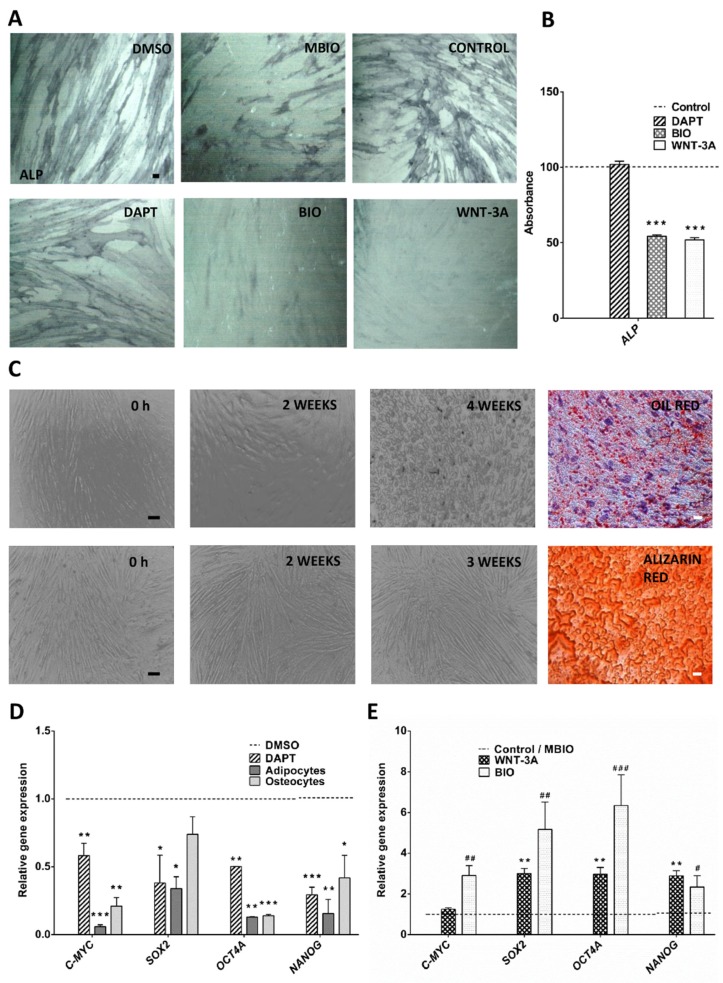
Notch and Wnt signaling regulate cell differentiation and pluripotency core factor expression in DPSCs (**A**)**:** ALP activity assay showed that Wnt activation suppressed the default osteoblastic pre-commitment in DPSCs. Scale bar = 100 µm. (**B**): Quantification of ALP absorbance in DPSC cultures after WNT-3A/BIO application (**C**): DPSC differentiation to adipocytes and osteocytes. Phase-Contrast (PC) Microscopy and Alizarin S Red and Oil Red staining showed a phenotypic change and terminal differentiation of control DPSCs after adipoinduction and osteoinduction treatments. Top panel: terminal adipocyte differentiation was assessed by Oil Red (bright red spots) staining after 4 weeks, cell nuclei are counterstained with Hematoxylin; bottom panel: terminal osteoblastic differentiation was assessed by Alizarin Red staining after 3 weeks. Scale bar = 100 µm (Alizarin, Oil Red). Scale bar = 20 µm (PC) (**D**): Q-PCR transcript expression analysis for core pluripotency factors C-MYC, SOX2, OCT4A and NANOG between control and terminally differentiated DPSC cultures, and also between control and DAPT-treated DPSCs (**E**)**:** Q-PCR analysis of core factors in DPSC cultures after BIO/WNT-3A application, with respect to their respective controls MBIO/PBS (dashed line). Data are normalized to reference *β-ACTIN* and *GAPDH* levels and presented as the mean+SEM (*n* = 3). *: *p* < 0.05; **: *p* < 0.01; ***: *p* < 0.001. Dunn’s Test, Kruskal- Wallis H Test. Asterisks (*) report significance with respect to controls PBS/DMSO, and Hash symbol (#) represents significance with respect to control MBIO. #: *p* < 0.05; ##: *p* < 0.01; ###: *p* < 0.001.

**Figure 2 cells-09-00652-f002:**
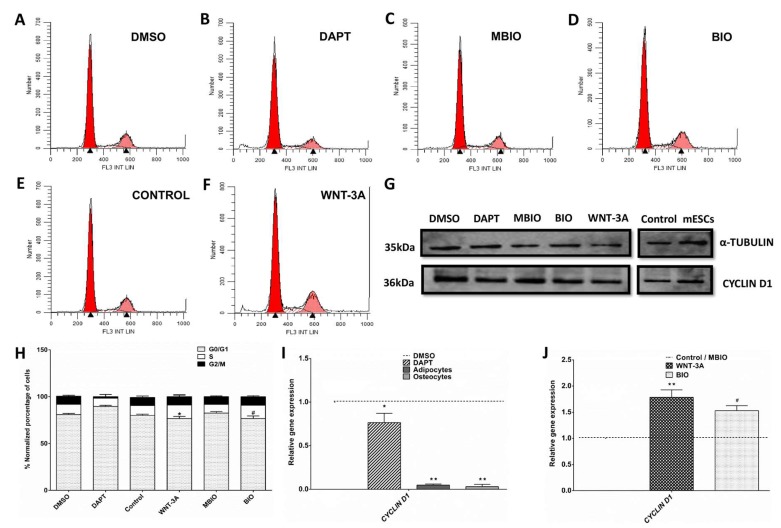
Notch/Wnt treatment effect over cell cycle in DPSCs. (**A**–**F**): Flow cytometry analysis of DPSC cultures exposed to different conditions for 48 h (DMSO, DAPT, MBIO, BIO, WNT-3A). Control (**E**) represents DPSCs grown in standard medium. (**G**): Western Blot showing CYCLIN D1 protein expression in the different conditions. (**H**): Representation of the percentage of cells in the different cell cycle phases (G0/G1, S, G2/M) under each condition. Data are presented as the mean+SEM (*n* = 8). (**I**) Q-PCR analysis of CYCLIN D1 transcript expression in DPSCs exposed to DAPT or terminal adipogenic and osteogenic cell differentiation treatments, with respect to controls (dashed line). (**J**) Q-PCR analysis of CYCLIN D1 transcript expression in BIO and WNT-3A-treated DPSCs with respect to their respective controls MBIO/PBS (dashed line). Data are normalized to reference β-ACTIN and GAPDH levels and presented as the mean + SEM (*n* = 3). *: *p* < 0.05; **: *p* < 0.01; ***: *p* < 0.001. Dunn’s Test, Kruskal- Wallis H Test. Asterisks (*) report significance with respect to controls PBS/DMSO, and Hash symbol (#) represents significance with respect to control MBIO.

**Figure 3 cells-09-00652-f003:**
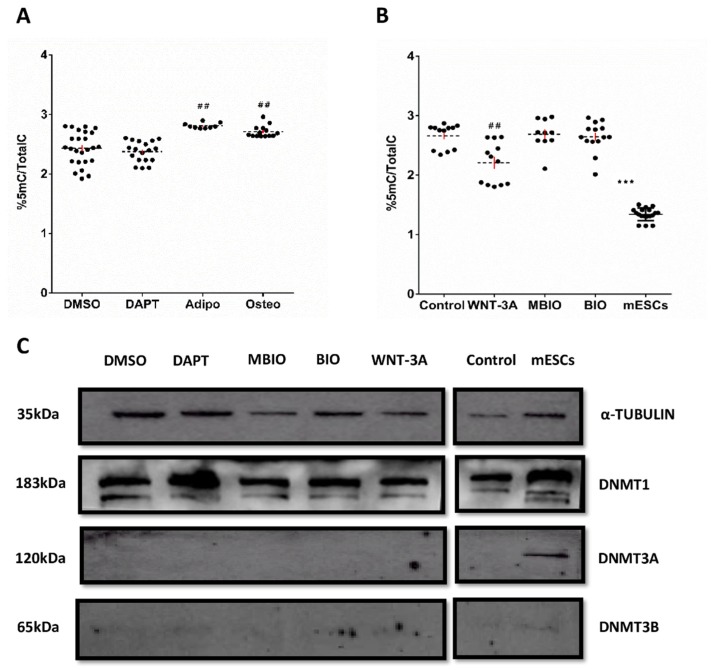
Notch and Wnt signaling regulate genomic DNA methylation in DPSCs. (**A**,**B**): Global DNA methylation levels in DPSCs showing the proportion (%) of 5 mC with respect to total C in control (DMSO, MBIO) with respect to DAPT, BIO, WNT-3A and differentiation induction conditions. Mouse ESCs were used as a control for low DNA methylation. Data are presented as the mean+SEM *(n* = 8). ##: *p* < 0.01, ***: *p* < 0.001; Dunn’s post-hoc test, Kruskal- Wallis H Test. (**C**): Representative WB showing DNMT1, DNMT3A and DNMT3A expression in DPSCs under different treatments. α-TUBULIN was used as a protein loading control. Mouse ESCs were used as a positive control for DNMT protein expression (right panel). Control on right panel shows untreated DPSCs grown in standard conditions.

**Figure 4 cells-09-00652-f004:**
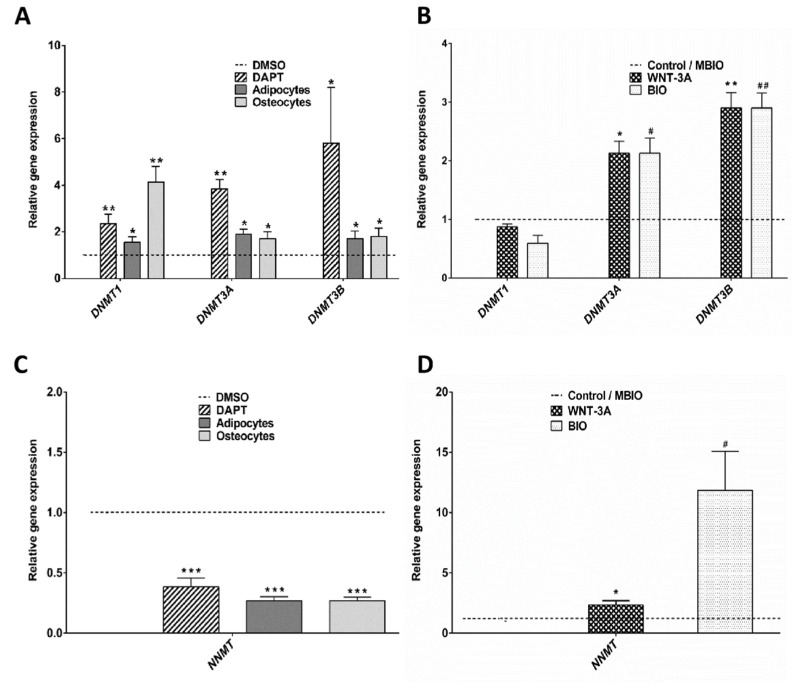
Notch and Wnt signaling affect the expression of methyltransferases in DPSCs. (**A**,**B**): Q-PCR showing relative differences in DNA-methyltransferases DNMT1, DNMT3A and DNMT3B expression in DPSCs between control (DMSO, MBIO) and DAPT, BIO, WNT-3A, adipoinduction and osteoinduction conditions. Data are normalized to reference β-ACTIN and GAPDH levels and represented as the mean + SEM (*n* = 3). (**C,D**): Q-PCR analysis showing relative differences in transcript expression for the Nicotinamide-N-methyltransferase NMMT in DPSCs subjected to different treatments. Data are normalized to reference β-ACTIN and GAPDH levels and represented as the mean+SEM (*n* = 3) *: *p* < 0.05; **: *p* < 0.01; ***: *p* < 0.001. Dunn’s Test, Kruskal- Wallis H Test. Asterisks (*) report significance with respect to controls PBS/DMSO, and Hash symbol (#) represents significance with respect to control MBIO. #: *p* < 0.05; ##: *p* < 0.01.

**Figure 5 cells-09-00652-f005:**
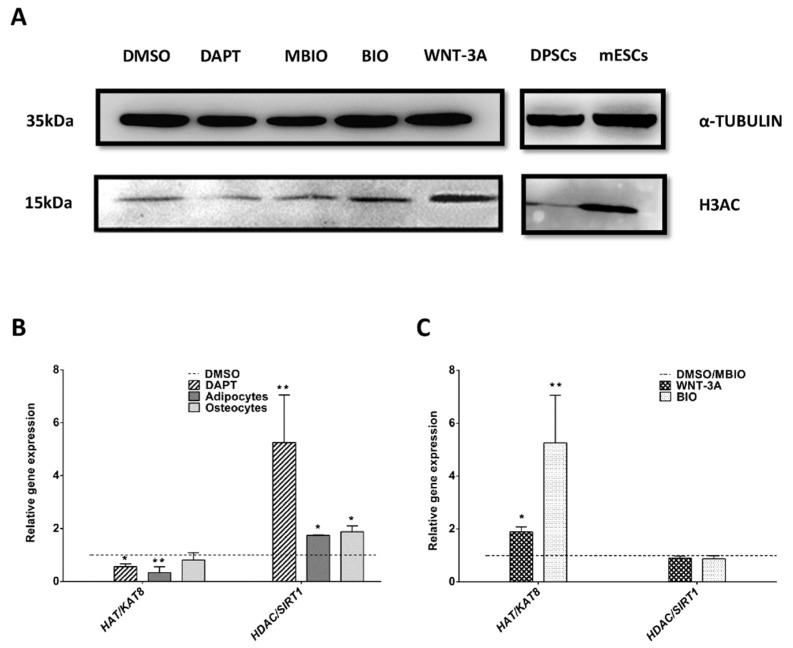
Notch and Wnt signaling affect histone acetylation in DPSCs. (**A**): Representative WB showing H3AC levels in control and treated DPSCs. α-TUBULIN was used as a protein loading control. Mouse ESCs were used as a positive control for high levels of H3 acetylation (right panel). Control on right panel shows untreated DPSCs grown in standard conditions. (**B**): Q-PCR analysis of HAT/KAT8 and HDAC/SIRT1 expression in DAPT-treated and osteo/adipoinduced DPSC cultures. (**C**): Q-PCR analysis of WNT-3A/BIO treated DPSC cultures for HAT/KAT8 and HDAC/SIRT1. Data are normalized to reference β-ACTIN and GAPDH levels and presented as the mean + SEM (*n* = 3). *: *p* < 0.05; **: *p* < 0.01. Dunn’s Test, Kruskal- Wallis H Test. Asterisks (*) report significance with respect to controls PBS/DMSO, and Hash symbol (#) represents significance with respect to control MBIO.

**Figure 6 cells-09-00652-f006:**
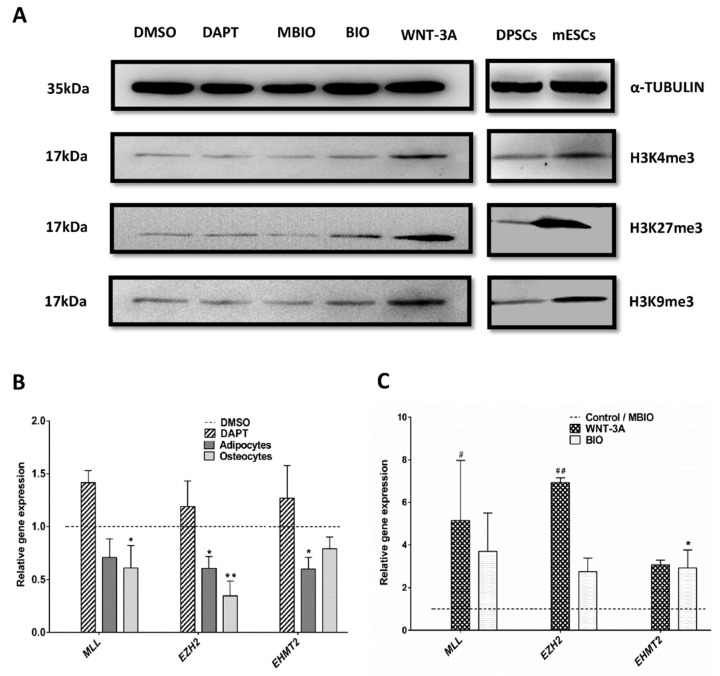
Wnt activation regulates H3 trimethylation tags in DPSCs. (**A**): Representative WBs showing protein levels of trimethylated H3K4me3, H3K27me3, and H3K9me3. α-TUBULIN was used as a protein loading control. Mouse ESCs were used as a positive control for high H3 methylation (right panel). Control on right panel shows untreated DPSCs grown in standard conditions. (**B**,**C**): Q-PCR analyses showing transcript expression levels for histone-methyltransferases placing H3K4me3 (MLL), H3K27me3 (EZH2) and H3K9me3 (EHMT2) epigenetic tags. Data are normalized to reference β-ACTIN and GAPDH levels and represented as the mean+SEM (*n* = 3). *: *p* < 0.05; **: *p* < 0.01; ***: *p* < 0.001. Dunn’s Test, Kruskal- Wallis H Test. Asterisks (*) report significance with respect to controls PBS/DMSO, and Hash symbol (#) represents significance with respect to control MBIO. #: *p* < 0.05; ##: *p* < 0.01.

**Table 1 cells-09-00652-t001:** Primer pairs to assess gene transcript expression in DPSCs by qPCR.

Primers		Sequence 5′–3′	Annealing (ºC)	Amplicon (bp)
*β-ACTIN*	Upstream	GTTGTCGACGACGAGCG	58.5	93
	Downstream	GCACAGAGCCTCGCCTT	59.7	
*GAPDH*	Upstream	CTTTTGCGTCGCCAG	60.3	131
	Downstream	TTGATGGCAACAATATCCAC	60.8	
*DNMT1*	Upstream	CGTAAAGAAGAATTATCCGAGG	60.5	123
	Downstream	GTTTTCTAGACGTCCATTCAC	57.7	
*DNMT3A*	Upstream	GAAGAGAAGAATCCCTACAAAG	57.6	136
	Downstream	CAATAATCTCCTTGACCTTGG	60	
*DNMT3B*	Upstream	CTTACCTTACCATCGACCTC	57.7	167
	Downstream	ATCCTGATACTCTGAACTGTC	54.7	
*NNMT*	Upstream	CTGACTACTCAGACCAGAAC	53.6	113
	Downstream	TCTGTTCCCTTCAAGATCAC	59.3	
*KAT8/HAC*	Upstream	GAAATATGAGAAGAGCTACCG	57.2	123
	Downstream	ATCTTATGGTCTTTGCCATC	58	
*SIRT1/HDAC*	Upstream	AAGGAAAACTACTTCGCAAC	57.6	89
	Downstream	GGAACCATGACACTGAATTATC	59.7	
*OCT4A*	Upstream	CGTGAAGCTGGAGAAGGAGA	60.7	137
	Downstream	CATCGGCCTGTGTATATCCC	60.1	
*CMYC*	Upstream	GTCAAGAGGCGAACACACAAC	59.8	162
	Downstream	TGGACGGACAGGATGTATGC	59.8	
*NANOG*	Upstream	GTCAAGAAACAGAAGACCAG	56.4	184
	Downstream	GCCACCTCTTAGATTTCATTC	59.2	
*SOX2*	Upstream	ATAATAACAATCATCGGCGG	61.1	90
	Downstream	AAAAAGAGAGAGGCAAACTG	57.8	
*CYCLIN D1*	Upstream	TGAGGCGGTAGTAGGACAGG	60.4	140
	Downstream	GACCTTCGTTGCCCTCTGT	59.6	
*EZH2*	Upstream	CCAACACAAGTCATCC	60.4	91
	Downstream	CCATAAAATTCTGCTGTAGGG	59.6	
*MLL*	Upstream	AAAGACTTCTAAGGAGGCAG	61.1	183
	Downstream	AACATATAGCAACCAATGCC	58.7	
*EHMT2*	Upstream	CTGTCAGAGGAGTTAGGTTC	60.4	135
	Downstream	ATCCACAGAGTAGGAATCATAG	59.6	
*PPARγ*	Upstream	AAAGAAGCCAACACTAAACC	57.4	78
	Downstream	TGGTCATTTCGTTAAAGGC	60.1	
*LPL*	Upstream	ACACAGAGGTAGATATTGGAG	54.8	143
	Downstream	CTTTTTCTGAGTCTCTCCTG	55.7	
*SPARC*	Upstream	CTTCAGACTGCCCGGAGA	61.1	90
	Downstream	GAAAGAAGATCCAGGCCCTC	60.2	
*OSTERIX*	Upstream	TGAGGAGGAAGTTCACTATG	53.8	200
	Downstream	CATTAGTGCTTGTAAAGGGG	54.0	

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
