# Peer review of "Wnt-3a Induces Epigenetic Remodeling in Human Dental Pulp Stem Cells"

_cells, 2020, doi:10.3390/cells9030652_

Round 1
Reviewer 1 Report
The manuscript is very interesting. Although this, there are many concerns that need to be addressed.
The authors should quantify the ALP. Oil Red O and Alizarin Red s are aspecific and very bad. The authors should change the images. Moreover, the authors should test osteogenesis and adipogenesis related markers. They should demonstrate that stemness markers increase and , in parallel, differentiation markers decrease.
Cell cycle analyses are very bad. Usually Modfit software is used to evaluate cell cycle. Coefficient of variation is too high. Therefore, the experiments of cell cycle must be re-performed. Cyclin D expression must be confirmed by western blotting or cytometry. Sometimes, the levels of mRNA do not correspond to corresponding protein levels. Also DNMT3A and 3B and DNMT1 levels must be confirmed by western blotting.
The Discussion section is too long, dispersive and disorganized. Please, modify it
Author Response
Dear reviewer
We are pleased to resubmit the updated version of our manuscript “WNT-3A induces epigenetic remodeling in Dental Pulp Stem Cells” (Uribe-Etxebarria et al.), which has been modified according to your suggestions.
New experimental work has been added to this version to address the raised comments, which were all meaningful and useful. New author Igor Irastorza was also added to the new author’s list for his contribution to these results. The new manuscript version contains now 6 Figures instead of 5, plus a supplementary one.
We hope you appreciate the improvements made to the manuscript.
We proceed now to give one-by-one responses to each and every of the raised comments and suggestions:
Reviewer #1
1- The manuscript is very interesting. Although this, there are many concerns that need to be addressed.The authors should quantify the ALP. Oil Red O and Alizarin Red s are aspecific and very bad. The authors should change the images. Moreover, the authors should test osteogenesis and adipogenesis related markers. They should demonstrate that stemness markers increase and , in parallel, differentiation markers decrease.
We are thankful to the reviewer for his/her comments. We have quantified ALP absorbance and included the data in figure 1 (new Fig.1B). The results confirm a significant decrase in ALP staining after Wnt activation. Oil Red and Alizarin Red images have been replaced with new ones, showing better the lipid droplets and calcified matrix precipitates. We have also included new data of osteogenesis markers (SPARC, OSTERIX) and adipogenesis markers (PPARγ, LPL) expression. We found that those differentiation markers rised when DPSCs were subjected to terminal differentiation conditions, and decreased when DPSCs were subjected to Wnt activation by either BIO or WNT-3A for 48h. All these results are included in a new Supplementary Figure 1.
2- Cell cycle analyses are very bad. Usually Modfit software is used to evaluate cell cycle. Coefficient of variation is too high. Therefore, the experiments of cell cycle must be re-performed. Cyclin D expression must be confirmed by western blotting or cytometry. Sometimes, the levels of mRNA do not correspond to corresponding protein levels. Also DNMT3A and 3B and DNMT1 levels must be confirmed by western blotting.
We have used Modfit software to reanalyze our flow cytometry data. Figure 2 has been completely rearranged to show the results more clearly. We have also performed WB to assess Cyclin D expression (also included in the new figure). Despite the observed changes by qPCR, the changes in CyclinD expression did not translate to the protein level.
We have also performed WB for the DNA methyltransferases DNMT1, DNMT3A and DNMT3B, after purchasing the corresponding primary antibodies (which took a pretty long time to arrive to the lab). The results show that DNMT1 is the principal DNA methyltransferase expressed by DPSCs. DNMT3A and DNMT3B could not be detected by WB in DPSCs in any of the conditions, which confirmed their very low levels of expression. However, a small DNMT3A and DNMT3B expression could still be detected in DPSCs at transcript level. Probably this marginal transcript expression was not enough to generate easily detectable protein bands by WB, despite their upregulation at 48h. Of note, DNMT3A and DNMT3B expression could also be detected by WB for the positive control of mESCs, but even there the protein detection levels were not particularly high, especially in the case of DNMT3B. We have included WB results in the new Figure 3, and in order not to overcharge the previous figure plate, we have decided to split it into two parts: New Figure 3, showing the MS and WB data, and New Figure 4, showing the corresponding qPCR data.
Regarding DNMT1, its expression at both transcript and protein levels in DPSCs does not seem to be altered by the Wnt activation treatments. DNMT1 protein levels were unaffected by BIO/WNT-3A, as assessed by WB. We added new qPCR experiments to confirm this and plotting all the recent and previous data together the decrease by qPCR was no longer statistically significant (although quite close; p~0.06). In view of these results we no longer think the observed decrease in DNA methylation after WNT-3A addition observed by MS could be explained by a downregulation of DNMT1 expression, and we are now more inclined to consider other mechanisms of regulation of DNMT1 enzymatic activity (by NMMT overexpression, histone acetylation, etc.). The corresponding parts on the results and the discussion sections have been modified to account for this.
3- The Discussion section is too long, dispersive and disorganized. Please, modify it
We have substantially shortened the number of words and paragraphs in the discussion section and tried to summarize and clarify better our line of thinking.
Reviewer 2 Report
General comments:
In this paper, the authors investigated the effects of human recombinant protein WNT-3A on DNA methylation and histone acetylation in human dental pulp stem cells. Their results showed that a 48h exposure to the WNT-3A changed the epigenetic profile of DPSCs, by decreasing DNA methylation and increasing histone acetylation. This is an interesting study. The authors did well on performing their experiments. However they need to be careful about their statement. My concerns about this review are as below.
Author Response
Dear reviewer,
We are pleased to resubmit the updated version of our manuscript “WNT-3A induces epigenetic remodeling in Dental Pulp Stem Cells” (Uribe-Etxebarria et al.), which has been modified according to your suggestions.
New experimental work has been added to this version to address the raised comments, which were all meaningful and useful. New author Igor Irastorza was also added to the new author’s list for his contribution to these results. The new manuscript version contains now 6 Figures instead of 5, plus a supplementary one.
We hope you appreciate the improvements made to the manuscript.
We proceed now to give one-by-one responses to each and every of the raised comments and suggestions:
Reviewer #2
1- In this paper, the authors investigated the effects of human recombinant protein WNT-3A on DNA methylation and histone acetylation in human dental pulp stem cells. Their results showed that a 48h exposure to the WNT-3A changed the epigenetic profile of DPSCs, by decreasing DNA methylation and increasing histone acetylation. This is an interesting study. The authors did well on performing their experiments. However they need to be careful about their statement.
We are thankful to the reviewer for his/her positive consideration of this study. We agree that the word “reprogramming” was probably not the best to describe this effect. We did not mean a full cell reprogramming, but rather a partial one (i.e. increased stemness without reaching pluripotency). In order to provide a more accurate statement of our findings, we have decided to replace the word “reprogramming” with “remodeling” in the manuscript title.
2- The upregulated gene expression doesn’t mean that wnt-3s-treated DPSCs become pluripotent or multipotent. The authors should check that if wnt-3a-treated DPSCs are able to differentiate into multiple lineages, such as neuronal, myocardial and pancreatic by in vitro differentiation. They also can check if wnt-3a-treated DPSCs form teratoma by in vivo transplantation.
As commented, we did not intend to claim that DPSCs became pluripotent after Wnt activation. We understand that all these experiments would be very pertinent in that case.
3- The authors need to test the effects of Notch/Wnt treatment on cell viability in DPSCs. Why did the authors choose 48 hr for treatment?
The time of 48h was selected because this is the one we used in previous studies and we already knew this treatment exerted a significant effect on DPSC stemness and metabolism (Uribe-Etxebarria et al. 2017 Eur Cell Mater.; Uribe-Etxebarria et al. 2019 J Cell Phys.). In our previous study (Uribe-Etxebarria et al. 2017 Eur Cell Mater; PMID : 29092089) we report that these treatments have no deletereous impact on DPSC viability or apoptosis, as assessed by Calcein/Propidium Iodide staining and WB for active PARP.
4- Does Notch/Wnt treatment have similar effects on epigenetic changes in other types of cells (for example, fibroblasts from skin) ?
To our knowledge, no equivalent experiments to ours have been performed using other cell types, although there are interesting publications showing a link between Notch/Wnt activity, epigenetic regulation and cell proliferation and/or differentiation. In the case of skin fibroblasts, the hyperactivation of Wnt signaling and excessive cell proliferation in fibroblasts from systemic sclerosis patients has been shown to be dependent on DNA methylation, and DNMT inhibition reduces Wnt activity in those skin fibroblasts (Dees et al. 2014 Ann. Rheum. Dis; PMID: 23698475). As another example with a notable clinical application, Notch/Wnt pharmacological regulators in combination with HDAC inhibitors are now on their way to approval as a treatment to paliate hearing loss associated to aging. There is evidence suggesting that this combinatorial treatment might induce the transdifferentiation of choclear supporting cells into mechanosensory hair cells, which degenerate with natural aging (Smarajeeva et al. 2019: Mol. Ther. PMID: 30982678). Therefore, in view of the evidence presented in these particular types of cells, combined with our own results with dental pulp cells, it is indeed very possible that Notch/Wnt pathways make part of a global epigenetic regulatory network in many cell types in the human body.